A new species of Psychrophrynella (Amphibia, Anura, Craugastoridae) from the humid montane forests of Cusco, eastern slopes of the Peruvian Andes

Catenazzi Alessandro 1 2 acatenazzi@gmail.com
Ttito Alex 3
1 Department of Zoology, Southern Illinois University Carbondale , Carbondale IL , United States
2 Centro de Ornitología y Biodiversidad , Lima , Peru
3 Museo de Historia Natural, Universidad Nacional de San Antonio Abad , Cusco , Peru
Hughes Jane
Electronic publication date: 2016 Mar 14
Publication date: 2016
Volume: 4
Electronic Location ID: e1807
Received 2016 Jan 17; Accepted 2016 Feb 23
Copyright: ©2016 Catenazzi and Ttito
Copyright year: 2016
Copyright holder: Catenazzi and Ttito
License: This is an open access article distributed under the terms of the Creative Commons Attribution License, which permits unrestricted use, distribution, reproduction and adaptation in any medium and for any purpose provided that it is properly attributed. For attribution, the original author(s), title, publication source (PeerJ) and either DOI or URL of the article must be cited.
License URL: https://creativecommons.org/licenses/by/4.0/

Keywords: Cloud forest, Psychrophrynella chirihampatu, Chytrid fungus, Bioacoustics, Frog, Leaf litter amphibian, Paucartambo

Funding: Mohamed bin Zayed Species Conservation Fund Disney Worldwide Conservation Fund Rufford Small Grants Foundation Southern Illinois University This research was supported by grants from the Mohamed bin Zayed Species Conservation Fund, the Disney Worldwide Conservation Fund, the Rufford Small Grants Foundation and Southern Illinois University startup funds to AC. The funders had no role in study design, data collection and analysis, decision to publish, or preparation of the manuscript.

==============================
We describe a new species of Psychrophrynella from the humid montane forest of the Department Cusco in Peru. Specimens were collected at 2,670–3,165 m elevation in the Área de Conservación Privada Ukumari Llakta, Japumayo valley, near Comunidad Campesina de Japu, in the province of Paucartambo. The new species is readily distinguished from all other species of Psychrophrynella but P. bagrecito and P. usurpator by possessing a tubercle on the inner edge of the tarsus, and from these two species by its yellow ventral coloration on abdomen and limbs. Furthermore, the new species is like P. bagrecito and P. usurpator in having an advertisement call composed of multiple notes, whereas other species of Psychrophrynella whose calls are known have a pulsed call (P. teqta) or a short, tonal call composed of a single note. The new species has a snout-vent length of 16.1–24.1 mm in males and 23.3–27.7 mm in females. Like other recently described species in the genus, this new Psychrophrynella inhabits high-elevation forests in the tropical Andes and likely has a restricted geographic distribution.

Introduction

The frog genus Psychrophrynella currently includes 21 species distributed across the humid grasslands and forests from 1830 to 4190 m.a.s.l. in the Amazonian slopes of the Andes in southern Peru and Bolivia (De la Riva & Burrowes, 2014; Duellman & Lehr, 2009, Frost, 2015). The genus was placed within the Holoadeninae in the family Strabomantidae by Hedges, Duellman & Heinicke (2008), but Pyron & Wiens (2011) synonymized Strabomantidae with Craugastoridae. Only three species are currently known from Peru, but most of the eastern valleys of the Andes in the southern Peruvian regions of Cusco and Puno have been poorly explored and are likely to contain many undescribed species (Catenazzi & Von May, 2014).

The phylogenetic relationships among the Holoadeninae genera Noblella and Psychrophrynella are not fully resolved. The type species of Psychrophrynella, P. bagrecito (Lynch, 1986) is found in the upper watershed of the Araza river in the Peruvian region of Cusco (Lynch, 1986). Despite having been chosen as the type species for the genus by Hedges, Duellman & Heinicke (2008), P. bagrecito possess several morphological traits that are shared with some species of Noblella, rather than with species of Psychrophrynella (De la Riva, Chaparro & Padial, 2008a; Lehr, 2006). Furthermore, the type species of Noblella, N. peruviana (Noble, 1921) is only known from three type specimens collected from 1899 to 1900 at a Peruvian locality in Region Puno (Noble, 1921), and some distinctive traits such as the presence of tubercles might be difficult to discern in long preserved specimens (De la Riva, Chaparro & Padial, 2008b). Finally, P. bagrecito, P. usurpator, N. lochites, and possibly N. peruviana, according to the original description (Noble, 1921), share the unique trait among congeneric species of possessing an elongated tarsal fold.

Surveys in the humid montane forests of the Japumayo Valley in the Region of Cusco, Peru, recently revealed the existence of a species of Psychrophrynella with an elongated tarsal fold, yellow ventral coloration and a long advertisement call composed of multiple notes, unlike known congeneric species. Here we describe this new species, and we report on surveys of infection with the pathogenic fungus Batrachochytrium dendrobatidis in populations of the new species and of sympatric amphibians. This fungus has caused the collapse of amphibian biodiversity in humid montane forests of the Tropical Andes (Catenazzi et al., 2011; Catenazzi, Lehr & Vredenburg, 2014), and could threaten amphibians at the type locality of the new species.

Methods

The format of the diagnosis and description follows Duellman & Lehr (2009) and Lynch & Duellman (1997), except that the term dentigerous processes of vomers is used instead of vomerine odontophores (Duellman, Lehr & Venegas, 2006). Taxonomy follows Hedges, Duellman & Heinicke (2008) except for family placement (Pyron & Wiens, 2011). Meristic traits of similar species were derived from specimens examined, published photographs, or species descriptions (Table 1).

Table 1 Selected characters and character conditions in species of Psychrophrynella.

Selected characters (+ = character present; − = character absent) and character conditions among Bolivian (first column) and Peruvian (all other columns) species of Psychrophrynella.

Characters	Bolivian spp.	P. bagrecitoa	P. boettgerib	P. chirihampatu	P. usurpatorc	
Maximum SVL (mm)	19.0–30.9	19.0	18.4	27.7	30.5	
Tympanic membrane	Not differentiated	Not differentiated	Distinct	Not differentiated	Not differentiated	
Vomerine teeth	–	–	–	–	–	
Dorsolateral folds	Variable	Weak, anterior only	+	Weak, anterior only	Weak, anterior only	
Vocal sac	Variable	+	?	+	+	
Vocal slits	Variable	+	–	+	+	
Nuptial pads	–	–	–	–	–	
Finger I vs. II	Variable	Shorter	Shorter	Shorter	Slightly shorter or equal	
Inner tarsal tubercle	–	Sickle-shaped	–	Elongated	Elongated	
Ventral coloration	Variable	White, brown marks	Brown and cream	Yellow, brown flecks	Brown or tan, cream flecks	
Call	Single noted	Multiple notese	?	Multiple notes	Multiple notese	
Notes.

a Sample size for SVL is 17 individuals measured from 1999 to 2009 (unpublished data).

b Source: Lehr (2006).

c Sample size for SVL is 811 individuals measured from 1996 to 2015 (unpublished data).

d Except for P. saltator and P. taqta.

e Unpublished data.

Specimens were preserved in 70% ethanol. Sex and maturity of specimens were determined by observing sexual characters and gonads through dissections. We measured the following variables (Table 2) to the nearest 0.1 mm with digital calipers under a stereomicroscope: snout–vent length (SVL), tibia length (TL), foot length (FL, distance from proximal margin of inner metatarsal tubercle to tip of Toe IV), head length (HL, from angle of jaw to tip of snout), head width (HW, at level of angle of jaw), eye diameter (ED), tympanum diameter (TY), interorbital distance (IOD), upper eyelid width (EW), internarial distance (IND), and eye–nostril distance (E–N, straight line distance between anterior corner of orbit and posterior margin of external nares). Fingers and toes are numbered preaxially to postaxially from I–IV and I–V respectively. We determined comparative lengths of toes III and V by adpressing both toes against Toe IV; lengths of fingers I and II were determined by adpressing these fingers against each other.

Table 2 Measurements of type series of Psychrophrynella chirihampatu.

Range and average (± standard deviation) measurements (in mm) of type series of Psychrophrynella chirihampatu sp. n.

Characters	Females (n = 10)	Males (n = 17)	
SVL	23.9–25.8 (25.0 ± 0.6)	16.1–21.7 (19.3 ± 1.6)	
TL	10.2–11.0 (10.8 ± 0.2)	8.0–10.1 (9.0 ± 0.6)	
FL	10.3–11.5 (11.0 ± 0.4)	7.1–10.4 (9.3 ± 0.8)	
HL	8.0–9.0 (8.5 ± 0.3)	6.3–8.1 (7.3 ± 0.5)	
HW	7.4–7.8 (7.6 ± 0.4)	5.6–7.9 (6.8 ± 0.6)	
ED	2.5–2.8 (2.7 ± 0.1)	2.0–2.5 (2.2 ± 0.2)	
IOD	2.5–2.8 (2.7 ± 0.1)	1.8–2.4 (2.1 ± 0.2)	
EW	1.6–2.0 (1.8 ± 0.1)	1.2–1.8 (1.4 ± 0.2)	
IND	2.3–2.7 (2.5 ± 0.1)	1.8–2.2 (2.0 ± 0.1)	
E–N	2.0–2.3 (2.1 ± 0.1)	1.5–2.0 (1.7 ± 0.1)	
TL/SVL	0.42–0.44	0.43–0.52	
FL/SVL	0.42–0.46	0.39–0.54	
HL/SVL	0.35–0.38	0.34–0.40	
HW/SVL	0.32–0.36	0.33–0.38	
HW/HL	0.91–0.98	0.88–1.07	
E–N/ED	0.71–0.85	0.68–0.90	
EW/IOD	0.64–0.71	0.58–0.75	

We performed Principal Component Analysis on morphological measurements for the new species and for the morphologically similar Psychrophrynella usurpator (Table 3). We retained five variables to maximize sample size of n = 17 for the new species and of n = 42 for P. usurpator. Morphometric data (non-transformed, after checking for normality) were analyzed with the princomp function using eigen on the correlation matrix in the ‘stats’ package in R 3.1.3 (The R Foundation for Statistical Computing; http://www.R-project.org). Principal Components 1 and 2 (representing 87% of variation) were used to produce a scatter plot. Proportion data were arcsine square root transformed for univariate comparisons. Variation in coloration was described on the basis of field notes and photographs of live frogs. Photographs taken by A. Catenazzi of live specimens, including types and non-collected specimens, and of preserved types have been deposited at the Calphoto online database (http://calphotos.berkeley.edu).

Table 3 Results from the Principal Component Analysis of 5 meristic characters (SVL, head length, head width, tibia length, foot length) of male adults of two populations of Psychrophrynella.

The highest loading for each component is in boldface.

Component	PC1	PC2	PC3	PC4	PC5	
Loadings						
Snout–vent length	0.49	−0.22	−0.01	0.00	−0.84	
Tibia length	0.46	0.38	−0.05	−0.79	0.17	
Foot length	0.44	0.45	−0.54	0.53	0.15	
Head length	0.45	0.09	0.80	0.31	0.23	
Head width	0.39	−0.77	−0.25	−0.03	0.43	
Importance of components						
Standard deviation	1.93	0.81	0.57	0.45	0.32	
Proportion of variance	0.74	0.13	0.07	0.04	0.02	
Cumulative proportion	0.74	0.87	0.94	0.98	1.00	

We recorded advertisement calls of male CORBIDI 16495 at the type locality on 21 June 2015 and recorded air temperature with a quick reading thermometer (recording #9843 deposited at the Fonoteca Zoológica, Museo Nacional de Ciencias Naturales, Madrid, www.fonozoo.org). We used a digital recorder (Zoom H2, recording at 48 kHz, 24-bit, WAV format) for field recording, and Raven Pro version 1.4 (Cornell Laboratory of Ornithology, Ithaca, NY) to analyze call variables. We analyzed a total of 26 calls. The following variables were measured from oscillograms: note and duration and rate, interval between notes or calls, number of pulses, and presence of amplitude modulation. Variables measured from spectrograms included dominant frequency, and presence of frequency modulation or harmonics. Spectral parameters were calculated through fast Fourier transform (FFT) set at a length of 512 points (Hann window, 50% overlap). Averages are reported ± SD.

We estimated genetic distances to confirm generic placement of the new species within Psychrophrynella through analysis of the non-coding 16S rRNA mitochondrial fragment. We used tissues from the holotype, CORBIDI 16495, and from paratopotype MHNC 14664, to obtain DNA sequences for the new species (deposited in GenBank; Appendix 1). We downloaded sequences of congeneric species and of Holoadeninae species in related genera (Barycholos, Bryophryne, Holoaden and Noblella) from GenBank (Appendix 1). Extraction, amplification, and sequencing of DNA followed standard protocols (Hedges, Duellman & Heinicke, 2008). We used the 16Sar (forward) primer (5′–3′ sequence: CGCCTGTTTATCAAAAACAT) and the 16Sbr (reverse) primer (5′–3′ sequence: CCGGTCTGAACTCAGATCACGT) (Palumbi et al., 2002). We employed the following thermocycling conditions during the polymerase chain reaction (PCR) with a Veriti thermal cycler (Applied Biosystems, Foster City, CA, USA): 1 cycle of 96 °C/3 min; 35 cycles of 95 °C/30 s, 55 °C/45 s, 72 °C/1.5 min; 1 cycle 72 °C/7 min. PCR products were purified with Exosap-IT (Affymetrix, Santa Clara, CA, USA) and shipped to MCLAB (San Francisco, CA) for sequencing. We used Geneious R8, version 8.1.6 (Biomatters, http://www.geneious.com/) to align the sequences with the MAFFT, version 7.017 alignment program (Katoh & Standley, 2013). We estimated uncorrected p-distances (i.e., the proportion of nucleotide sites at which any two sequences are different) with the R package “ape” (Paradis, Claude & Strimmer, 2004).

We swabbed specimens in the field to quantify infection by Batrachochytrium dendrobatidis (Bd). Each animal was swabbed with a synthetic dry swab (Medical Wire & Equipment, #113) using a standardized swabbing protocol. In post-metamorphic stages, swabs were stroked across the skin a total of 30 times: 5 strokes on each side of the abdominal midline, 5 strokes on the inner thighs of each hind leg, and 5 strokes on the foot webbing of each hind leg (total of 30 strokes/frog). We used a real-time Polymerase Chain Reaction (PCR) assay on material collected on swabs to quantify the level of infection (Boyle et al., 2004). DNA was extracted from swabs using PrepMan Ultra and extracts were analyzed in a Life Technologies StepOne Plus qPCR instrument following the protocol outlined in Hyatt et al. (2007) and Boyle et al. (2004), except that extracts were analyzed once (Kriger, Hero & Ashton, 2006). We calculated ZE, the genomic equivalent for Bd zoospores by comparing the qPCR results to a set of standards, and considered any sample with ZE >1 to be infected or Bd-positive.

Specimens examined are listed in Appendix 2; codes of collections are: CORBIDI = Herpetology Collection, Centro de Ornitología y Biodiversidad, Lima, Peru; MHNC = Museo de Historia Natural del Cusco; KU = Natural History Museum, University of Kansas, Lawrence, Kansas, USA; MUSM = Museo de Historia Natural Universidad Nacional Mayor de San Marcos, Lima, Peru; and MHNG = Muséum d’Histoire Naturelle, Genève, Switzerland.

Research was approved by Institutional Animal Care and Use Committees of Southern Illinois University Carbondale (protocols #13-027). Permit to carry on this research has been issued by the Peruvian Ministry of Agriculture (permit #292-2014-MINAGRI-DGFFS-DGEFFS). The Comunidad Campesina Japu Q’eros authorized work on their land.

The electronic version of this article in Portable Document Format (PDF) will represent a published work according to the International Commission on Zoological Nomenclature (ICZN), and hence the new names contained in the electronic version are effectively published under that Code from the electronic edition alone. This published work and the nomenclatural acts it contains have been registered in ZooBank, the online registration system for the ICZN. The ZooBank LSIDs (Life Science Identifiers) can be resolved and the associated information viewed through any standard web browser by appending the LSID to the prefix http://zoobank.org/. The LSID for this publication is: urn:lsid:zoobank.org:pub:34FC0393-6723-4554-912A-AEA7ED811589. The online version of this work is archived and available from the following digital repositories: PeerJ, PubMed Central and CLOCKSS.

Results

Psychrophrynella chirihampatu sp. n. urn:lsid:zoobank.org:pub:34FC0393-6723-4554-912A-AEA7ED811589.

http://zoobank.org/34FC0393-6723-4554-912A-AEA7ED811589.

Holotype (Figures 1–3, Table 2). CORBIDI 16495, an adult male (Figs. 2 and 3) from 13°26′44.92″S; 71°0′12.35″W (WGS84), 2,730 m.a.s.l., Área de Conservación Privada (ACP) Ukumari Llaqta, Comunidad Campesina de Japu, Distrito Paucartambo, Provincia Paucartambo, Región Cusco, Peru, collected by A. Catenazzi and A. Ttito on 21 June 2015.

Paratopotypes (Figure 4, Table 2) Ten total: five adult males, CORBIDI 16496 and 16497 and MHNC 14658, 14664 and 14666 (Figs. 2 and 3), and five adult females, CORBIDI 16498–16499, 16696 and MHNC 14661–14662, collected at the type locality by A. Catenazzi and A. Ttito on 21 June 2015.

Figure 4 Dorsolateral and ventral views of four paratypes of Psychrophrynella chirihampatu sp. n. showing variation in dorsal and ventral coloration.

Male MHNC 14656 (A, B), Tambo Japu. Male MHNC 14667 (C, D), type locality. Female CORBIDI 16502 (E, F), Playa camp site. Female CORBIDI 16499 (G, H), type locality. Photographs by A. Catenazzi.

Paratypes (Figure 4). 16 total, all from ACP Ukumari Llakta: nine adult males, CORBIDI 16505–16509 and MHNC 14656 and 14670–14672, and one adult female, CORBIDI 16504, collected near Tambo, 13°27′0.14″S; 71°02′11.40″W (WGS84), 3160 m.a.s.l., by A. Catenazzi and A. Ttito on 18 June 2015; two adult males, CORBIDI 16503 and MHNC 14667, and four adult females, CORBIDI 16501–2 and MHNC 14668–69, collected at Playa camp site, 13°26′53.52″S; 71°0′38.38″W (WGS84), 2780 m.a.s.l., by A. Catenazzi and A. Ttito on 18 June 2015.

Generic placement. A new species of Psychrophrynella as defined by (Duellman & Lehr, 2009; Hedges, Duellman & Heinicke, 2008). Frogs of the genus Psychrophrynella are morphologically similar and closely related to Barycholos, Bryophryne, Holoaden and Noblella (Hedges, Duellman & Heinicke, 2008; Heinicke, Duellman & Hedges, 2007; Padial, Grant & Frost, 2014). The new species is assigned to Psychrophrynella rather than any of the other genera on the basis of molecular data (Table 4) and overall morphological resemblance with the type species P. bagrecito (see Table 1), including presence of an elongated fold-like tubercle on the inner edge of tarsus. Genetic data confirm generic placement of the new species within Psychrophrynella. We found substantial genetic distances (uncorrected p-distances from 7.2–19.3%; Table 4) between P. chirihampatu and congeneric species for which mitochondrial sequence data were available. The most closely related species is P. usurpator (16S uncorrected p-distance: 7.2%), followed by P. guillei and P. wettsteini which had much higher distances of 17.8–19.3%. Species from other genera (with the exception of B. cophites) had genetic distances above 20%.

Figure 1 Map of Peru indicating the type localities of Peruvian and western Bolivian species of Psychrophrynella.

P. bagrecito (black square), P. boettgeri (black star), P. chirihampatu sp. n. (asterisk), P. guillei and P. saltator (white circle), P. kallawaya (white star), P. katantika (circle), and P. usurpator(triangle).

Figure 2 Photographs of live and preserved specimen of the holotype of Psychrophrynella chirihampatu.

Live (A, C, E) and preserved (B, D, F) specimen of the holotype of Psychrophrynella chirihampatu sp. n., male CORBIDI 16495 (SVL 18.8 mm) in dorsolateral (A, B), dorsal (C, D) and ventral (E, F) views. Photographs by A. Catenazzi.

Figure 3 Palmar and plantar surfaces of the holotype of Psychrophrynella chirihampatu.

Ventral views of hand (A) and foot (B) of holotype, CORBIDI 16495 (hand length 4.6 mm, foot length 8.7 mm) of Psychrophrynella chirihampatu sp. n. Photographs by A. Catenazzi.

Table 4 Genetic distances from 16S data.

Genetic distances (uncorrected p-distances) estimated from the non-coding 16S rRNA mitochondrial fragment between Psychrophrynella chirihampatu and related taxa (in boldface the most closely related species) of the subfamily Holadeninae (Craugastoridae).

	Bryophryne bakersfield	Bryophryne bustamantei	Bryophryne cophites	Barycholos pulcher	Holoaden luederwaldti	Noblella lochites	Noblella myrmecoides	P.a guillei	P. usurpator	P. wettsteini	P. chirihampatu MHNC14664	P. chirihampatu (holotype)	
Bryophryne bakersfield	0												
Bryophryne bustamantei	0.06	0											
Bryophryne cophites	0.04	0.17	0										
Barycholos pulcher	0.21	0.30	0.29	0									
Holoaden luederwaldti	0.19	0.26	0.24	0.30	0								
Noblella lochites	0.23	0.29	0.26	0.27	0.25	0							
Noblella myrmecoides	0.17	0.30	0.28	0.27	0.29	0.22	0						
P. guillei	0.18	0.15	0.26	0.29	0.24	0.29	0.30	0					
P. usurpator	0.21	0.26	0.28	0.31	0.25	0.31	0.29	0.23	0				
P. wettsteini	0.20	0.23	0.24	0.31	0.23	0.29	0.28	0.14	0.23	0			
P. chirihampatu MHNC14664	0.20	0.22	0.19	0.25	0.20	0.21	0.18	0.18	0.07	0.19	0		
P. chirihampatu (holotype)	0.20	0.22	0.19	0.25	0.20	0.22	0.18	0.18	0.07	0.19	0.00	0	

Diagnosis. A species of Psychrophrynella characterized by (1) skin on dorsum finely shagreen with some small warts forming linear ridges at mid dorsum; skin on venter smooth, discoidal fold not visible, thin dorsolateral folds visible on anterior half part of body; (2) tympanic membrane not differentiated, tympanic annulus barely visible below skin; (3) snout short, bluntly rounded in dorsal view and in profile; (4) upper eyelid lacking tubercles, narrower than IOD; cranial crests absent; (5) dentigerous process of vomers absent; (6) vocal slits present; nuptial pads absent; (7) Finger I shorter than Finger II; tips of digits bulbous, not expanded laterally; (8) fingers lacking lateral fringes; (9) ulnar tubercles absent; (10) heel lacking tubercles; inner edge of tarsus bearing an elongate, obliquous fold-like tubercle; (11) inner metatarsal tubercle prominent, elliptical, of higher relief and about one and a half times the size of ovoid, outer metatarsal tubercle; supernumerary plantar tubercles absent; (12) toes lacking lateral fringes; webbing absent; Toe V slightly shorter than or about the same length as Toe III; tips of digits not expanded, weakly pointed; (13) dorsum tan to brown and gray with dark brown markings; some individuals with a yellow or orange middorsal line extending from tip of snout to cloaca and to posterior surface of thighs; interorbital bar present; chest, venter and ventral parts of arms and legs yellow with brown flecks; throat and palmar and plantar surfaces brown or reddish-brown; (14) SVL 16.1–24.1 in males (n = 34), 23.3–27.7 in females (n = 12).

Comparisons. The new species differs from most described species in the genus by possessing an elongate fold-like tubercle on the inner edge of tarsus. Among currently known species in the genus, only the two Peruvian, and geographically closest species P. bagrecito and P. usurpator possess such a tubercle, which is similarly shaped (obliquous) in the latter but sickle-shaped in P. bagrecito. The other Peruvian species, P. boettgeri, and all species described from Bolivia (including P. guillei, P. katantika, P. kallawaya and P. saltator known from the Cordillera de Aplobamba near the border with Peru; Fig. 1) lack a tubercle or fold on the inner edge of tarsus. Furthermore, among species whose advertisement calls is known, P. chirihampatu shares with P. bagrecito, P. saltator and P. usurpator the characteristic of having a call composed of multiple notes (Table 1; unpublished data for calls of P. bagrecito and P. usurpator), whereas the call is pulsed in P. teqta or composed of short, single notes in other congeneric species (De la Riva, 2007; De la Riva & Burrowes, 2014).

Morphologically, the new species is most similar to P. usurpator (characters in parentheses; Table 1), from which it differs by having yellow ventral coloration with reddish-brown or grey flecks (dull brown, gray or black with cream flecks), Finger I shorter than Finger II (slightly shorter or same length), smaller SVL reaching 27.5 mm in females (SVL up to 30.5 mm), slender head (wider and shorter head), and inner metatarsal tubercle at least three times the size of outer metatarsal tubercle (about same size). The scatterplot of the first two Principal Components axes reveal that these two species occupy distinct regions of morphospace (Fig. 5A). Snout-vent length and tibia length load strongly on the first Principal Component axis PC1, whereas head width and foot length load strongly on the second Principal Component, PC2 (Table 3). Univariate comparisons of measurements of male P. chirihampatu and P. usurpator reveal that male P. chirihampatu have narrower heads, averaging 35.4% of SVL (HW 38.0% of SVL in P. usurpator; t57 = − 5.12, p < 0.001; Fig. 5B), and longer tibia length, averaging 46.7% of SVL (TL 45.2% of SVL, t57 = 2.24, p = 0.01), but no difference in foot length (t57 = 1.44, p = 0.08).

Figure 5 Morphometric comparisons between Psychrophrynella chirihampatu and P. usurpator.

(A) Principal components analysis of 5 meristic characters, and (B) relationship between head width and snout-vent length of 17 adult males of Psychrophrynella chirihampatu from the type locality and of 44 males P. usurpator from Abra Acjanaco, Manu National Park, Peru.

We also compared the new species with the type species of Psychrophrynella, P. bagrecito (Lynch, 1986). Psychrophrynella chirihampatu differs from P. bagrecito (characters in parentheses; Table 1) in having an elongated and oblique fold-like tarsal tubercle (short and sickle-shaped), broad dark markings on dorsum (longitudinal stripes), venter yellow with dark flecks (venter orange brown with light gray flecks) and larger size of females up to 27.5 mm in SVL (SVL of females up to 19.0 mm).

Ten other small species of craugastorid frogs of the subfamily Holoadeninae are known to occur in montane forests and high Andean grasslands south of the Apurimac canyon in Peru: Bryophryne abramalagae, B. bustamantei, B. cophites, B. flammiventris, B. gymnotis, B. hanssaueri, B. nubilosus, B. zonalis, Noblella madreselva and N. pygmaea. None of these species has the unique ventral coloration of P. chirihampatu, and all but B. gymnotis and the two species of Noblella (which are much smaller in size) lack a visible tympanic annulus.

Description of holotype. Adult male (18.8 mm SVL); head narrower than body, its length 39.9% of SVL; head slightly longer than wide, head length 110.3% of head width; head width 36.2% of SVL; snout short, bluntly rounded in dorsal and lateral views (Fig. 2), eye diameter 26.7% of head length, its diameter 1.1 times as large as its distance from the nostril; nostrils not protuberant, close to snout, directed laterally; canthus rostralis slightly concave in dorsal view, convex in profile; loreal region flat; lips rounded; upper eyelids without tubercles; upper eyelid width 59.1% of interorbital distance; interorbital region flat, lacking cranial crests; eye-nostril distance 90% of eye diameter; supratympanic fold weak; tympanic membrane not differentiated, tympanic annulus visible below skin; two small postrictal ridges on each side of head. Vocal sac and vocal slits present. Choanae round, small, positioned far anterior and laterally, widely separated from each other; dentigerous processes of vomers and vomerine teeth absent; tongue large, ovoid, not notched.

Skin on dorsum smooth with minute, scattered tubercles, denser posteriorly; barely visible dorsolateral folds anteriorly; skin on flanks and venter smooth; no pectoral fold, barely visible discoidal fold; cloaca not protuberant, cloacal region without tubercles. Ulnar tubercles and folds absent; palmar tubercle flat and oval, approximately same length but twice the width of elongate, thenar tubercle; supernumerary palmar tubercles absent; subarticular tubercles prominent, ovoid in ventral view, rounded in lateral view, largest at base of fingers; fingers lacking lateral fringes; relative lengths of fingers 3 >4 >2 >1 (Fig. 3); tips of digits bulbous, not expanded laterally (Fig. 3); forearm lacking tubercles.

Hindlimbs moderately long, tibia length 46.8% of SVL; foot length 46.3% of SVL; upper and posterior surfaces of hindlimbs smooth with scattered, minute tubercles; heel without tubercles; inner edge of tarsus bearing an elongated, oblique fold-like tubercle, outer edge of tarsus lacking tubercles; inner metatarsal tubercle, oval, high, and at least three times the size of conical, rounded outer metatarsal tubercle; few, minute plantar supernumerary tubercles weakly defined; subarticular tubercles rounded, ovoid in ventral view; toes lacking lateral fringes, not webbed; toe tips weakly pointed, not expanded laterally; relative lengths of toes 4 >3 >5 >2 >1 (Fig. 3); foot length 46.3% of SVL.

Measurements of holotype (in mm): SVL 18.8, TL 8.8, FL 8.7, HL 7.5, HW 6.8, ED 2.0, IOD 2.2, EW 1.3, IND 1.8, E–N 1.8.

Coloration of holotype in alcohol. Dorsal surfaces of head, body, and limbs grayish tan, with a dark brown X-shaped middorsal mark. The interorbital bar is a narrow dark stripe and is bordered anteriorly by a cream stripe. There is a dark brown subocular mark bordered by a thin cream line. A dark brown stripe, outlined below by a thin cream line extends from the tip of the snout to above the insertion of forelimb; from that point, a discontinuous dark line runs dorsolaterally separating dorsum from flank to the point of hind limb insertion. The iris is dark gray. The throat has brown coloration anteriorly, fading into pale grey with brown flecks posteriorly. This pale grey coloration extends from chest to belly, but turns to yellow posteriorly and on the ventral parts of hind and forelimbs. The posterior surfaces of thighs are dark brown with a narrow, pale gray stripe running diagonally from cloaca to inside of knee; the plantar and palmar surfaces are brown, but fingers and toes are cream. The dorsal surfaces of hind limbs have transverse dark bars.

Coloration of holotype in life. Similar to coloration in preservative, with the difference that the dorsal coloration is beige with red flecks, the cream stripes and lines on the head are bronze, the throat is reddish-brown with yellow flecks, the chest is yellow with reddish-brown flecks, the belly and ventral surfaces of hind and forelimbs are yellow, and the fingers and toes are reddish-brown at the base and yellow at the tip.

Variation. Coloration in life is based on field notes and photographs taken by A. Catenazzi of 23 collected and 21 uncollected specimens found at and near the type locality (Fig. 4, see Appendix S1 for codes and photographsof uncollected specimens). There is substantial variation in dorsal coloration, which varies from beige to grayish-tan and dark brown, and while most individuals have the X-shaped dorsal mark (barely noticeable in individuals with dark coloration), several individuals have additional dark marks. The dark stripe extending dorsolaterally between the points of insertion of limbs is discontinuous in most individuals (including the holotype) and absent in at least three specimens (CORBIDI 16496, 16504, and MHNC 14658), but at least ten specimens (CORBIDI 16497, 16499, 16506, MHNC 14668, 14671–72, and uncollected 639.15, 640.15, 1019.15, 10676.15) have a continuous stripe separating the lighter dorsal coloration from the darker coloration on the flanks. Sixteen individuals (36%; including paratypes CORBIDI 16496–98, 16503–06, 16993–94, MHNC 14667, 14670, 14672; and uncollected individuals 640.15, 1005.15, 1006.15, 1065.15) have a yellow or orange middorsal line extending in most individuals from the interorbital bar (but from tip of snout in CORBIDI 16496, 16993–94, and uncollected 1065.15) to the cloaca, and from the cloaca along the posterior side of thighs to the knee. The throat is generally reddish-brown with yellow or orange flecks; CORBIDI 16992, 16496 and uncollected 1065.15 have a yellow or orange line running midventrally from the tip of snout to the cloaca. Chest and ventral surfaces of abdomen and limbs are yellow or orange with variable amounts of reddish-brown, brown or grey flecks, especially on the chest. In some individuals (e.g., CORBIDI16504–06, 16994, and uncollected 1018.15) background coloration on chest and belly is brown or gray with yellow flecks.

The summary of measurements of all types is reported in Table 2. A histogram of the frequency distribution of SVL for all captured specimens (types and uncollected specimens) suggests modes of 20.0–21.9 mm for males and 24.0–25.9 mm for females (Fig. 6).

Figure 6 Frequency distribution of snout-vent lengths of Psychrophrynella chirihampatu.

Sample size is 23 types and 21 uncollected individuals of Psychrophrynella chirihampatu sp. n.

Figure 7 Advertisement call of Psychrophrynella chirihampatu.

Advertisement call of male CORBIDI 16495 (SVL 18.8 mm), holotype of Psychrophrynella chirihampatu sp. n., recorded at the type locality on 21 June 2015 (Tair = 11.6 °C).

Advertisement call. The call of the holotype was recorded shortly before capture at 13h45 on 21 June 2015 (Fig. 7). At a Tair = 11.6 °C, the advertisement call averaged 3,212 ± 1,005 ms in duration (range 1,140–4.524 ms) and consisted of 47.9 ±16.1 single-pulsed notes (range 10–68) produced at a rate of 14.7 ±1.8 notes/s (range 8.77–16.55). Peak frequency averaged 2,712 ± 33 Hz (range 2,584–2,885 Hz) and increased during calls (t1,78 = − 6.53, p < 0.01): peak frequency averaged 2,702 ± 38 Hz for the first three notes, and 2,748 ± 50 Hz for the last three notes of each call. Amplitude also increased during each call, and the three final notes had amplitude ∼400% higher than the amplitude of the three initial notes. Average note duration was 5.4 ± 1.2 ms (range 1–12 ms), but note duration increased from 2.6 ± 0.7 ms in the first three notes to 7.8 ± 1.3 ms in the last three notes of each call. Furthermore, call structure varied during a sequence of 26 calls produced at a rate of 9.43 calls/minute: the number of notes increased from 57 notes in the first two calls to 68 notes in the 5th call, and then progressively declined to 10 notes in the 26th call.

Figure 8 Habitat and egg nest of Psychrophrynella chirihampatu.

Collection localities of Psychrophrynella chirihampatu sp. n. in the upper Japumayo valley (A; view from lookout at 3,000 m): frogs were found under mosses and rocks along the trail at 3,160 m (B), and under rocks in a natural landslide at the type locality at 2,700 m (C), including an unattended nest under a rock (D; 10¢ coin is 20.5 mm in diameter). Photographs by A. Catenazzi.

Etymology. The name of the new species is a combination of Quechua words used in apposition meaning “toad” (“hampa’tu”) that lives in the “cold” (“chiri”). The name is a wordplay built upon the genus and species names sharing the same meaning of “frog inhabiting cold environments,” because the generic name Psychrophrynella derives from the Greek psychros (cold) and phrynos (toad).

Distribution, natural history and threats. The new species was found during amphibian surveys in the Japumato valley (Fig. 8A) conducted from 17 to 24 June 2015. We searched for frogs under rocks, logs, in the leaf litter and the understory along the transition from montane forest to high-Andean grassland (wet puna) from 2,650 to 4,600 m. Specimens of P. chirihampatu were only found at elevations from 2,650 to 3,180 m. Most specimens were found under rocks (many males were calling) during the day in areas of disturbed montane forest vegetation, such as the sides of the trail near the Tambo camp site (Fig. 8B), and natural landslides at the type locality (Fig. 8C) and at the Playa campsite. Field notes indicate that males were heard calling in similarly disturbed areas of the montane forest and along the edges of forest bordering landslides and other open areas.

We found an unattended nest of 11 eggs (Fig. 8D), diameter 4.5 mm on average, under a rock at the type locality. Ten female paratypes had 9.6 ± 1.5 eggs (range 7–12 eggs) at different stages of maturation; of these, one had 10 mature eggs averaging 3.9 ± 0.4 mm in diameter (range 3.5–4.6 mm).

None of the 45 specimens of P. chirihampatu tested for Bd were infected. Similarly, two sympatric species, Bryophryne zonalis (n = 6) and Gastrotheca cf. excubitor (n = 10) were Bd-negative, as were Bryophryne sp. (n = 4) from 3,820–3,050 m and an individual of Pleurodema marmoratum from 4,600 m.

The upper Japumayo valley is part of the Área de Conservación Privada Ukumari Llaqta, a protected area recognized by Peruvian environmental ministerial decree (No 301-2011-MINAM) in December 2011, and owned by the Comunidad Campesina Japu Q’eros. Therefore, the known distribution range of the species is protected. Although the valley is used for agricultural purposes, current land use appears to be sustainable and is unlikely to negatively affect populations of P. chirihampatu. Given this species’ affinity for disturbed areas, it is even possible that the current anthropogenic use of the montane forest might enhance the distribution of P. chirihampatu.

The current conservation status of P. chirihampatu is unknown. The populations we surveyed in the Japumayo valley were relatively large: for example at the type locality we found 25 frogs in 7 person-hours. We did not observe any direct threat to these populations during our visit. In absence of more detailed data regarding its extent of occurrence, and according to the IUCN Red List criteria and categories (IUCN, 2013), this species can provisionally be considered for the “Data Deficient” category of the Red List.

Discussion

The new species is yet another addition to the ever growing list of small craugastorid frogs (genera Bryophryne, Noblella and Psychrophrynella) from the eastern slopes of the Peruvian and Bolivian Andes (Catenazzi, Uscapi & Von May, 2015; De la Riva, 2007; De la Riva & Burrowes, 2014; De la Riva, Chaparro & Padial, 2008a; Harvey et al., 2013; Lehr & Catenazzi, 2008; Lehr & Catenazzi, 2009a; Lehr & Catenazzi, 2009b; Lehr & Catenazzi, 2010). Most if not all of these species have narrow distribution ranges often restricted to the type locality and surrounding mountaintop region, although large areas in between the type localities of these species remain unexplored. It is remarkable however that mountain passes separated by less than 50 km in airline distance do not share any species of Bryophryne, Noblella or Psychrophrynella. Such high levels of observed beta diversity, and the presence of unexplored regions suggest that more species remain to be discovered.

We assign the new species to Psychrophrynella on the basis of shared meristic traits, general body shape and appearance, and overall similarity with the type species P. bagrecito and with P. usurpator. Interestingly, these two species shares with P. chirihampatu characters that are absent in other congeneric forms, such as the presence of an elongated tubercle on the inner edge of the tarsus, and an advertisement call composed of multiple notes. Further work is needed to document variation in meristic traits and acoustic properties of advertisement calls in species of Psychrophrynella, as well as molecular analyses aimed at determining the phylogenetic relationships of these species.

Although we did not detect the presence of Bd in the Japumayo valley, this fungus has been reported from the nearby region of Abra Huallahualla and Coline (approximately 15–20 km SW by airline from the type locality of P. chirihampatu), where infected species included terrestrial-breeding B. zonalis, and aquatic-breeding P. marmoratum and Telmatobius marmoratus (Catenazzi et al., 2011). Furthermore, members of the Japu Q’eros Community who guided us to the type locality confirmed that T. marmoratus, a species known to be susceptible to chytridiomycosis (Catenazzi & Von May, 2014; Warne et al., 2016), was previously abundant in the upper reach of the Japumayo valley, but had disappeared sometime during the last decade. Therefore, it is likely that Bd has already reached, and possibly caused declines of other amphibian populations in the Japumayo valley. In the montane forests of Manu NP (70 km NW of Japumayo), Bd has caused the local extinction of many stream-breeding species, but not of terrestrial-breeding frogs such as Psychrophrynella species (Catenazzi et al., 2011). These findings suggest that Bd might not be as much of a threat for P. chirihampatu as it is for aquatic-breeding frogs.

Species with narrow geographic distributions are intrinsically threatened, and they are less likely to be included in nationally protected areas, as previously shown for Peru (Catenazzi & Von May, 2014; Von May et al., 2008). Smaller areas, but more widely dispersed in the landscaspe, are needed to protect amphibian biodiversity in regions of high beta diversity such as tropical Andean mountaintops. The introduction of new legal forms of protected areas in Peru, such as conservation concessions, private and communal reserves, could greatly benefit amphibian conservation. Discovery of endemic species provides justification for these reserves; for example, the description of P. chirihampatu for the Área de Conservación Privada Ukumari Llakta means that this reserve now protects at least one species of amphibian not found anywhere else. Exploration of other private protected areas and conservation concessions will generate similarly beneficial outcomes and will advance our knowledge of amphibian biodiversity.

Supplemental Information

Appendix S1 Photographs of all types and uncollected specimens

Click here for additional data file.

Data S1 Raw data and catalogue entries for collected and uncollected specimens

Click here for additional data file.

We thank the Comunidad Campesina Japu Q’eros for their hospitability, granting us access to and guiding us through the Japumayo valley; and the Asociación para la Conservación de la Cuenca Amazónica for logistical support, and especially Marlene Mamani for introducing us to the community and for coordinating our visit to Japu. We thank I. De la Riva and M.P. Heinicke for comments on the manuscript.

Appendix 1

Gene sequences for molecular analyses

Genbank accession numbers for the taxa and genes sampled in this study.

Taxon	Voucher Nbr.	16S	
	
Barycholos pulcher	KU 217781	EU186709	
Bryophryne bakersfield	MHNC 5999	KT276289	
Bryophryne bustamantei	MHNC 6019	KT276293	
Bryophryne cophites	KU 173497	F493537	
Holoaden luederwaldti	MZUSP 131872	EU186710	
Noblella lochites	KU 177356	EU186699	
Noblella myrmecoides	QCAZ 40180	JX267542	
Pyschrophrynella guillei	AMNH-A 165108	AY843720	
Pyschrophrynella usurpator	KU 173495	F493714	
Pyschrophrynella wettsteini	KU 183049	EU186696	
Psychrophrynella chirihampatu	MHNC 14664	KU884560	
Psychrophrynella chirihampatu	CORBIDI 16495	KU884559	

Appendix 2

Specimens examined

Noblella madreselva (2 specimens): PERU: Cusco: Provincia La Convención, Madre Selva (Santa Ana), CORBIDI 15769–70.

Noblella pygmaea (15 specimens): PERU: Cusco: Provincia Paucartambo, Kosñipata, MHNG 2725.29–30, MUSM 24535–36, 26306–7, 26318–20, 30423–24, 30453–54, MTD 47286–87.

Psychrophrynella bagrecito (14 specimens): PERU: Cusco: Quispicanchis: Marcapata, Río Marcapata, below Marcapata, ca. 2740 m, KU 196512 (holotype), KU 196513–18, 196520–21, 196523–25 (all paratypes); La Convención: Hacienda Huyro between Huayopata and Quillabamba, 1830 m, KU 196527–28.

Psychrophrynella usurpator (78 specimens): PERU: Cusco: Provincia Paucartambo, Kosñipata, MUSM 20011, 20873–81, 20896–20913, 20925–33, 20946–47, 20955–57, 21012–18, 26272–73, 26278–79, 26308, 27592, 27906, 27950, 28033–28047, 30303, 30305, 30396–30400, 30405–30409, 30471–30474.

Additional Information and Declarations

Competing Interests

Author Contributions

Animal Ethics

Field Study Permissions

Data Availability

New Species Registration

The authors declare there are no competing interests.

Alessandro Catenazzi conceived and designed the experiments, performed the experiments, analyzed the data, contributed reagents/materials/analysis tools, wrote the paper, prepared figures and/or tables, reviewed drafts of the paper.

Alex Ttito conceived and designed the experiments, performed the experiments, contributed reagents/materials/analysis tools, reviewed drafts of the paper.

The following information was supplied relating to ethical approvals (i.e., approving body and any reference numbers):

Institutional Animal Care and Use

Committees of Southern Illinois University Carbondale #13-027.

The following information was supplied relating to field study approvals (i.e., approving body and any reference numbers):

Peruvian Ministry of Agriculture #292-2014-MINAGRI-DGFFS-DGEFFS.

The following information was supplied regarding data availability:

Photographs: Calphotos (http://calphotos.berkeley.edu)—photos can be accessed by searching the database with the species name, or by downloading Appendix S1 (Supplemental Information, see below).

Recording: FonoZoo (www.fonozoo.org)—recording can be accessed by searching the database with the species name.

The following information was supplied regarding the registration of a newly described species:

Species LSID: http://zoobank.org/NomenclaturalActs/286E2FCE-4D10-4609-B295-DD4FD1EBE691

Publication LSID: urn:lsid:zoobank.org:pub:34FC0393-6723-4554-912A-AEA7ED811589.

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
