# Peer review of "A new species of Psychrophrynella (Amphibia, Anura, Craugastoridae) from the humid montane forests of Cusco, eastern slopes of the Peruvian Andes"

_PeerJ, doi:10.7717/peerj.1807_

## Round 0.1 · original submission · Minor Revisions

Please address all the reviewers comments in your revision. I agree with reviewer 2 that inclusion of some genetic data might be useful - but not mandatory.
All the other suggestions seem reasonable to me and should improve the paper.

·

Basic reporting

No comments

Experimental design

No comments

Validity of the findings

The manuscript describes a species of frog new to science, so this is an important finding.

Additional comments

This manuscript is clear and well written, and it represents a worthwhile contribution to the knowledge of the amphibian fauna of the Andes of southern Peru. I have made some minor comments (and corrections) in the attached PDF file (from the original Word document), which should be addressed by the authors in order to clarify some points.

·

Basic reporting

The study is very well-written, cites the major relevant background literature including both systematic background and descriptions of related species, and is structured appropriately for a species description.

The one suggestion I have is that the systematic background (lines 56-67) could be expanded a bit since these frogs are a taxonomically problematic group. As it is written now, readers may mistakenly get the impression that all species of Noblella and Psychrophrynella form a clade and need to be merged into a single genus, when they actually don't. The main issue is not phylogeny but rather taxonomy. There are clearly two phylogenetic lineages, but nomenclatural confusion exists especially because for many years specimens now described as P. usurpator were incorrectly assigned to Phrynopus (=Noblella) peruvianus, which has persisted to some extent in the literature even after De la Riva et al. fixed the problem in 2008 (fas in, for example, Pyron and Wiens 2011). It is correct, however, that if the type of Noblella peruviana actually belongs to the clade currently called "Psychrophrynella", then this clade would properly be called Noblella, and the valid name of the clade currently called "Noblella" would default to Phyllonastes. Within Psychrophrynella, it is also worth pointing out that Hedges et al. 2008 included the Peruvian species P. usurpator and the Bolivian species P. iatamasi and P. wettsteini, so although there are morphooogical differences between Peruvian and Bolivian species, at least some are known to belong to the same clade.

Experimental design

The experimental design follows that of a typical species description, and is formatted in the standard that has been used in related Craugastorid frogs for years. Methods are clearly described and would be easily reproduced. Relevant specimens on which measurements were made are properly referenced. It would be useful if a small amount of genetic data were included (such as a piece of the 16S gene) to confirm generic placement. However, given that the new species is morphologically similar to P. usurpator and comes from a locality fairly close to known localities of P. usurpator, I don't see inclusion of genetic data as mandatory. A few sentences or a citation on the diagnostic usefulness of color pattern to characterize species of Psychrophrynella or other craugastorids would also be welcome.

Validity of the findings

The overall findings are valid. The primary aim of a species description is to demonstrate the new species differs from known species with one or more diagnostic characters. The authors meet this aim by providing several diagnostic characters, including coloration and relative tubercle size. They also follow all other requirements of the Code of Zoological Nomenclature in producing a valid species description.

I do not find it necessary personally, but if PeeJ policy is to make all data publicly available, the raw measurements used in the PCA should be included in an appendix.

Additional comments

No comments beyond what is stated above.

---

## Round 0.2 · accepted · Accept

The authors have satisfactorily addressed the comments of reviewers, and have included genetic data.